# From Gene Expression to Tissue Morphology: Can Generative Models Uncover the Link?

**Frederieke Lohmann**  FLOHMANN@ETHZ.CH
**Alberto Valdeolivas**  ALBERTO.VALDEOLIVAS_URBELZ@ROCHE.COM
**Jelica Vasiljević**  JELICA.VASILJEVIC@ROCHE.COM
*Roche Pharma Research and Early Development*
*Data & Analytics*
*Roche Innovation Center Basel, Switzerland*

## Abstract

Spatial Transcriptomics technologies enable capturing gene expression within the native tissue context. Platforms such as 10x Visium and Visium HD integrate gene expression with histological imaging, providing a multi-dimensional view of tissue organisation. Motivated by the success of generative models in computer vision and natural language processing, we investigate the largely unexplored task of synthesising histological images directly from gene expression profiles. Leveraging recent advancements in Spatial Transcriptomics, particularly the 10X Visium HD platform, we introduce the first two-stage conditional generative framework to infer tissue morphology from near-whole transcriptome profiles. Competitive FID scores and a study involving multiple pathologists confirm that the synthesised images are plausible and that our framework generalises well to unseen standard Visium samples. Furthermore, model interpretation reveals connections between structurally relevant gene sets and specific morphological patterns, opening new avenues for studying the relationship between gene expression and tissue morphology.

**Keywords:** generative models, spatial transcriptomics, contrastive learning, digital pathology

## 1 Introduction

Spatial Transcriptomics (ST) is an innovative technology that allows the capture of gene expression (GEX) profiles within their natural tissue environment. Unlike bulk RNA sequencing, which provides average GEX across the entire processed sample, ST delivers spatially resolved molecular data, paving the way for detailed study of tissue diversity, cellular interactions and disease progression. In recent years, a variety of ST technologies have been developed (Hahn et al., 2025), some of which, such as Visium and VisiumHD, combine GEX data with histological images, enabling simultaneous analysis of molecular and morphological characteristics.

While many studies have focused on predicting GEX from tissue morphology (Nonchev et al., 2025; Pizurica et al., 2024; Xie et al., 2023), comparatively little attention has been paid to the inverse problem—understanding how GEX profiles shape morphological features. Yet, uncovering this relationship can be crucial for tasks such as predicting the structural impact of gene perturbations and detecting early morphological changes linked to disease onset. Considering the tight coupling between structure and function in biological systems, it can be hypothesised that certain aspects of tissue morphology are encoded within the GEX profile. However, this relationship is complex and indirect: GEX leads to protein production, but the final tissue structure and function are primarily governed by protein-level processes such as post-translational modifications and

protein-protein interactions. Consequently, a given GEX profile may correspond to multiple morphological patterns, as the path from GEX to tissue structure involves several intermediate steps and regulatory mechanisms. Inspired by the effectiveness of generative models in text-to-image synthesis, we developed a two-stage conditional generative framework that models this relationship as a GEX-conditional probability distribution over histological images. Our framework synthesises plausible histology-like images, confirmed by competitive FID scores and expert pathologist reviews. Interpretability analyses further point to associations between certain GEX patterns and distinct morphological traits, setting the stage for future work to identify genes whose disruption may contribute to disease or drug-induced toxicity. To the best of our knowledge, this is the first approach to integrate high-resolution ST and conditional generative modelling to predict histological images. The code and data are publicly available https://github.com/lohmannf/genes2morphology.

## 2 Related Work

The emergence of generative architectures such as generative adversarial networks (GANs) and denoising diffusion probabilistic models has greatly enhanced synthetic histological image generation (Rashidi et al., 2025). One particularly promising direction is conditional generation, where models are guided to produce images with specific characteristics (Dolezal et al., 2023), enabling the creation of synthetic data for rare morphological classes. While conditioning on discrete values such as tissue type or disease has been widely explored, fewer studies focus on continuous features like GEX profiles. The generation of histopathology images from global GEX profiles has been investigated by Carrillo-Perez et al. (2023), who introduced RNA-GAN, a model for generating H&E brain and lung image tiles conditioned on bulk RNA-seq data. However, since bulk RNA-seq obscures intra-sample heterogeneity by averaging GEX signals across the entire tissue section, their model cannot capture or generate distinct morphological regions within the tissue. As a result, RNA-GAN lacks the ability to control finer morphological details or reproduce region-specific variations in tissue structure. Similarly, Carrillo-Perez et al. (2024) proposed RNA-CDM, a cascaded diffusion model for generating H&E image tiles from five cancer types while conditioning on bulk RNA-seq data from whole tissue samples. RNA-CDM inherits the same limitation as RNA-GAN: bulk sequencing data provides generalized information averaged across the sample and lacks spatial information. Navidi et al. (2024) introduced MorphoDiff, a latent diffusion model designed for generating Cell Painting fluorescent microscopy image tiles conditioned on chemical or genetic perturbations. Their method encodes specific few-gene perturbations using the single-cell foundational model scGPT (Cui et al., 2024), providing a more nuanced approach to modelling cellular responses. With the rise of imaging-based ST technologies, such as Xenium and CosMx, new methods have emerged to condition the generation of immunofluorescence images (IF) on GEX data. Wu and Koelzer (2024) proposed SST-Editing, a method built on StyleGAN2 (Karras et al., 2020) to generate spatially resolved IF image patches. The model was trained on DAPI-stained image tiles from CosMx and Xenium datasets, focusing on individual cells, with corresponding GEX data injected as an additional style code. Furthermore, they developed an inverted GAN to enable bidirectional editing of real images, allowing transitions between tumor and non-tumor morphologies. This concept was extended in their follow-up work (Wu et al., 2023), where they scaled GEX-based editing from single cells to entire whole-slide images (WSIs) using Xenium data. However, the generation in both approaches remains limited to a small, targeted gene panel, making hypothesis-agnostic investigation of the link between GEX and morphology difficult (Hahn et al.,

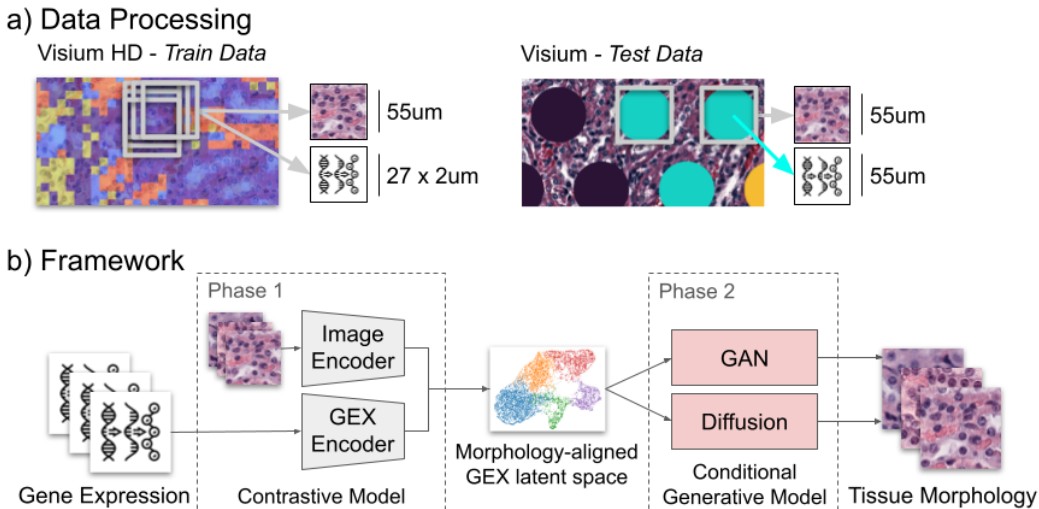

**Figure 1.** a) Data Processing strategy and b) two-phase modeling framework

2025). In contrast, sequencing-based ST technologies like Visium and Visium HD offer unbiased, near whole-transcriptome coverage but remain unexplored in the context of histological image generation. Visium HD, in particular, achieves sub-single-cell resolution with $2\mu m$ barcoded bins covering the tissue area without gaps, allowing precise spatial localization of transcripts. To the best of our knowledge, no existing work has specifically addressed the challenge of generating histological H&E images from near-whole transcriptome GEX data.

## 3 Method

In this work, we characterise the mapping from GEX profiles to tissue morphology as a one-to-many mapping and model it using a two-phase conditional generative approach. The workflow is depicted in Fig. 1b.

### 3.1 Phase 1: Contrastive Representation Learning

In the first step, we align tissue morphology and GEX by mapping them into a common representation space. The goal of this phase is dimensionality reduction of the GEX profiles to avoid overfitting in the second phase. While selecting highly variable genes (HVGs) or spatially variable genes (SVGs) is common practice, they have to be determined across all training and testing samples to be meaningful, limiting generalizability to new unseen samples. To mitigate this issue, we instead take a contrastive approach to dimensionality reduction. Inspired by prior work (Xie et al., 2023; Min et al., 2024; Lee et al., 2024), we train a bimodal contrastive model composed of an image encoder and a GEX encoder that integrates the two modalities into a shared latent space. The GEX encoder is a Multi-Layer Perceptron (MLP) with a projection head, while the image encoder combines the fixed pathology foundation model UNI (Chen et al., 2024) with a trainable projection head. The representations of matching data pairs are aligned using the CLIP loss (Radford et al., 2021).

## 3.2 Phase 2: Conditional Generation

In the second phase, the learned GEX representations are used to condition an image generative model. This phase is agnostic to the specific conditional generative architecture. We illustrate this flexibility by applying it to two major types of generative models: Generative Adversarial Networks (GANs) and diffusion models. For GANs, we adapt the StyleGAN-T architecture (Sauer et al., 2023) as StyleGAN-G, while for diffusion models, we use Stable Diffusion (SD) (Rombach et al., 2022). Both models are conditioned on the GEX embeddings obtained with the fixed GEX encoder pretrained in the first phase. The StyleGAN-T architecture is modified so that the discriminator incorporates a feature extractor trained on histological images instead of ImageNet (Dosovitskiy et al., 2020; Filiot et al., 2024), and it is trained from scratch. Similarly to StyleGAN-T, to ensure alignment between the GEX condition and the generated image, the spherical distance between the GEX embedding and the generated image embedding obtained with the pretrained image encoder is used as an additional guidance loss term. On the other hand, due to computational costs, the SD model was fine-tuned (U-Net part) on histological image data while conditioning on the GEX embeddings following the approach of Navidi et al. (2024). To mirror the usage of CLIP guidance in the GAN-based approach and improve prompt alignment, we employ classifier-free guidance (Ho and Salismans, 2022) with different strengths $w$ at inference time. Training details for both models are described in Appendix A.

## 3.3 Data

We conducted experiments on six formalin-fixed paraffin-embedded (FFPE) healthy mouse kidney samples, profiled using two spatial transcriptomics platforms: Visium HD (one sample) and standard Visium (five samples). Visium HD offers much higher spatial resolution, capturing gene expression in $2\mu m$ bins, compared to the $55\mu m$ spots of standard Visium. To leverage the high resolution of Visium HD samples, we aggregated adjacent $2\mu m$ bins into $\sim 300,000$ "pseudo-spots" by summing gene expression profiles over $27 \times 27$ $2\mu m$ bins with a stride of 5 bins. We retained 18,248 genes common to both platforms and normalised and log-transformed gene expression profiles per sample. To obtain GEX-morphology data pairs, the corresponding tissue morphology to each (pseudo-)spot is extracted from the Macenko stain-normalised (Macenko et al., 2009) WSI as a $55\mu m \times 55\mu m$ patch at the (pseudo-)spot's spatial location. We resize patches to $128 \times 128$ pixels to ensure uniform absolute resolution. The preprocessing workflow is illustrated in Fig. 1a. Given the need for a large training dataset, Visium HD data were used for model training, while standard Visium samples served as test data. Additional details on datasets and preprocessing are provided in Appendix B.

# 4 Results

We performed various experiments to quantitatively and qualitatively evaluate the plausibility of the generated images. Additionally, we analyzed the interpretability of the gene expression encoder and its relation to the biological relevance of the generated morphologies.

## 4.1 Image Plausibility

Figure 2 shows histological patches generated using multiple noise vectors (columns) and gene expression inputs (rows). The GEX profiles corresponding to real patches (shown in the leftmost

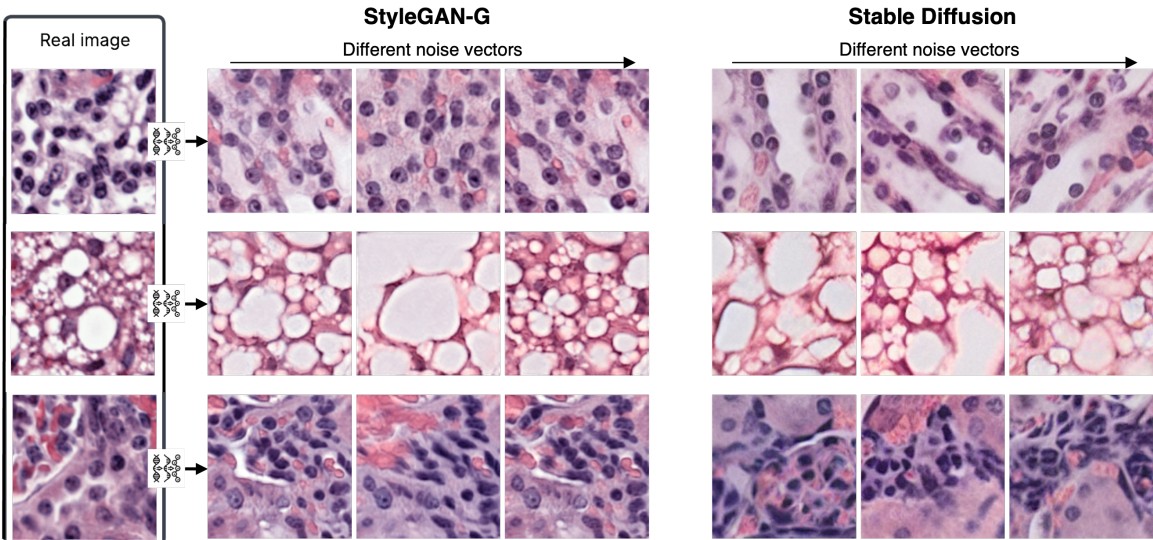

**Figure 2.** $128 \times 128$px images generated from different gene expression prompts with StyleGAN-G and Stable Diffusion (guidance strength $w = 3$). Corresponding reference image shown in the leftmost column. Columns correspond to the same random noise vector.

column) are used as input. The generated images accurately capture the tissue morphology of their corresponding reference images and can be visually compared to real tissue patches shown in Appendix C. Notably, the generated images more closely resemble the training data in hue and overall quality (Appendix C, top row), as the model does not explicitly account for experimental variations such as tissue preparation. To quantitatively measure the quality of generated images, Table 1 reports Fréchet Inception Distance (FID) (Heusel et al., 2017) and Kernel Inception Distance (KID) (Bińkowski et al., 2018) for both the GAN- and diffusion-based models. Since predicting tissue morphology from GEX using ST platforms such as Visium has not been previously addressed, there are no directly comparable methods. Therefore, we compare our results to the most closely related approaches. In this setting, StyleGAN-G achieves substantially better FID scores than prior GAN-based methods (e.g., RNA-GAN, FID = 83.89 (Carrillo-Perez et al., 2023)), while the diffusion-based model performs comparably to related diffusion methods (MorphoDiff, FID $\geq$ 78 (Navidi et al., 2024)). We attribute the relatively lower performance of SD-models in this setting to the limited generalizability of pretrained diffusion models to histopathology data, as noted in prior work

**Table 1.** FID and KID on 36528 generated images

| MODEL | FID $\downarrow$ | KID $\downarrow$ |
|---|---|---|
| STYLEGAN-G | **18.41** | **0.0127(0.0015)** |
| SD ($w = 1$) | 84.28 | 0.0666(0.0035) |
| SD ($w = 3$) | 78.99 | 0.0784(0.0043) |
| SD ($w = 7$) | 84.69 | 0.0925(0.0045) |

(Müller-Franzes et al., 2023). Due to the substantial training requirements of dedicated diffusion models and the limited availability of data, we selected the StyleGAN-G architecture for subsequent experiments. However, as more ST datasets become available, diffusion models specifically tailored to histopathology may outperform the GAN-based approach. To further assess the morphological plausibility of the generated images, we conducted an expert validation study with a panel of 10 veterinary pathologists. We presented each expert with a morphologically diverse, balanced dataset of 50 real and generated images and asked to classify them. The average classification performance across the dataset is shown in Tab. 2. The expert panel achieved a near-random accuracy of 0.486 on the entire dataset, confirming that the images generated with StyleGAN-G are highly realistic. In Appendix D, we compare the distributions of real and synthetic images in the learned image embedding space of the contrastive model as an additional evaluation of generated image quality.

### 4.2 GEX Latent Embedding Space

To assess the quality of the learned embedding space, we applied sample-wise Leiden clustering (Traag et al., 2019) on the GEX embedding space (GEX data projected using the pretrained gene expression encoder). For visualization, we combine all samples together and project them using UMAP (McInnes et al., 2018). The hyperparameters of the clustering are chosen to ensure the identification of key kidney regions within each sample, including the cortex (CX), outer stripe of the outer medulla (OSOM), brown adipose tissue (BAT), inner stripe of the outer medulla (ISOM), and inner medulla&papilla (IM&P) (Kumaran and Hanukoglu, 2024). Figure 3a+b presents the UMAP of the GEX embedding space. In Fig. 3a, embeddings are color-coded by Leiden clustering results, while Fig. 3b displays colors corresponding to sample origin. Figure 3d-h maps these clusters onto test set images, demonstrating successful identification of key kidney regions across all samples. The embedding space exhibits two important properties. GEX embeddings from different samples are well-integrated as demonstrated in Fig. 3b. Additionally, clusters representing identical tissue regions from various samples consistently localize to similar areas within the latent space, as shown in Fig. 3a. In contrast, in the UMAP space of the GEX data, a clear separation between samples can be observed (Fig. 3c). To quantify the alignment in the embedding space, we performed joint clustering across all test samples both in the GEX embedding and GEX space and compared it to sample-wise clustering and sample index using the Adjusted Rand Index (ARI). A visualization of the joint clustering can be found in Appendix E. The results show a strong alignment with separately identified clusters (ARI = 0.79) and minimal correlation with the sample index (ARI = 0.005) in the embedding space. In contrast, the overlap between clustering and sample index in GEX space is significant (ARI = 0.588), indicating stronger batch effects.

**Table 2.** Average expert classification performance on 50 images

| ACCURACY ↑ | PRECISION ↑ | RECALL ↑ |
|---|---|---|
| 0.486(0.095) | 0.492(0.096) | 0.488(0.111) |

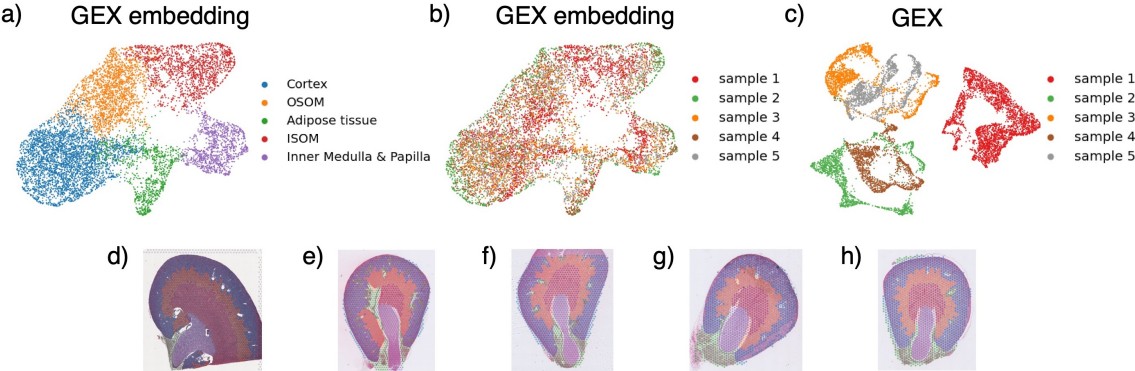

**Figure 3.** UMAP of the gene expression embedding space colored by (a) clusters found on each sample individually and (b) sample index. (c) UMAP of the gene expression space of samples 1-5 colored by sample index. Tissue cuts with spots colored by clusters found in gene expression embedding space for samples (d) 1, (e) 2, (f) 3, (g) 4, and (h) 5. Colors in (a) and (d)-(h) refer to the same clusters.

**Table 3.** Significantly enriched GO:CC terms in sample 4 with more than 4 leading genes.

| TERM | NES ↑ | QUERY | REFERENCE |
|---|---|---|---|
| LIPID DROPLET | 1.710422 | BAT | IM&P |
| BASOLATERAL PLASMA MEMBRANE | 1.702832 | CX | BAT |
| BASOLATERAL PLASMA MEMBRANE | 1.660759 | CX | OSOM |
| ENDOPLASMATIC RETICULUM LUMEN | 2.098712 | IM&P | OSOM |
| INTRACELLULAR ORGANELLE LUMEN | 2.002516 | IM&P | OSOM |
| COLLAGEN-CONTAINING EXTRACELLULAR MATRIX | 1.936724 | IM&P | OSOM |

## 4.3 Biological Relevance

Building on the observed alignment between GEX embedding space clusters and kidney regions (Fig. 3), we aim to identify which genes primarily drive the differences between clusters and therefore relate to the underlying morphological distinctions. We employed the Integrated Gradients (IG) method (Sundararajan et al., 2017), as implemented by Heimberg et al. (2024), to identify genes that contribute most significantly to differences between GEX embedding clusters. This method assigns an attribution score to each input feature (in this case, each gene) by quantifying its importance along a continuous path between two points in the embedding space. Therefore, we applied IG to assess gene contributions between pairs of cluster-averaged expression profiles as identified in Fig. 3d-h. This approach allowed us to determine which genes most significantly drive the differences between clusters in the gene expression embedding space. Our analysis involved the following steps: 1) We averaged the GEX embedding across all data points within each cluster. 2) For each pairwise comparison of cluster-averaged embeddings, we identified the 100 genes with the highest attribution scores. 3) We interpreted the attribution scores as indicators of a gene's influence on differences between clusters, analogous to differential gene expression analysis. We hypothesized that genes with high attribution scores would be associated with cellular components and structures that define kidney tissue architecture. To test this hypothesis, we performed Gene Set Enrichment Analysis (GSEA) Preranked (Subramanian et al., 2005) on these top 100 genes ranked by attribu-

tion score. We compared the most highly attributed genes to Gene Ontology Cellular Component (GO:CC) terms (Ashburner et al., 2000), as this ontology describes structural components rather than biological or molecular functions. We report enriched terms at a significance level of 2%. Table 3 illustrates the results for test sample 4, showing the normalized enrichment score (NES) of all significantly enriched GO:CC terms with at least 4 leading genes. The results for the remaining samples are provided in Appendix F. This representation highlights the most relevant structural components associated with the highly attributed genes in this sample.

Table 3 shows an enrichment of the Lipid Droplet term in brown adipose tissue as compared to other renal regions, which is expected given the high concentration of fat-storing adipocytes in this region. The embeddings of Basolateral Plasma Membrane genes distinguish the cortex from both the OSOM and brown adipose tissue. This reflects the cortex's abundance of proximal and distal convoluted tubules, whose extensive basolateral membranes support active secretion and absorption of solutes. In contrast, the OSOM has fewer of these tubule segments, while the brown adipose tissue is composed mainly of adipocytes which do not directly participate in solute transport. Consequently, basolateral membrane–related genes are more characteristic of the cortex's morphology than those of the OSOM or brown adipose tissue. Differences between the inner medulla/papilla and the OSOM emerge from terms linked to the extracellular matrix. The inner medulla and papilla contain specialized connective tissue that maintains structural integrity under the high osmotic pressure generated when urine is concentrated in the collecting ducts. In line with this function, renal interstitial cells in these regions produce collagen and other extracellular matrix components, which rationalizes the enrichment of the collagen-containing extracellular matrix term. Additionally, enrichment of genes associated with the endoplasmic reticulum (ER) lumen and intracellular organelle lumen further differentiates the inner medulla/papilla from the OSOM, reflecting higher levels of protein synthesis and post-translational modification. In particular, the collecting ducts—key structures within the inner medulla and papilla—rely on these processes to carry out their specialized functions.

The derived attribution scores offer an opportunity to identify novel gene sets that drive these morphological differences, providing a valuable direction for future research to pinpoint genes whose loss or alteration may contribute to disease or drug-associated toxicity.

## Conclusion

We introduce, to the best of our knowledge, the first approach to synthesise histology images directly from high-resolution GEX data. Experiments on five kidney samples demonstrate that our model learns biologically meaningful gene expression representations, generates histologically accurate tissue morphologies across diverse renal structures, and produces images that achieve competitive FID scores and expert pathologists find indistinguishable from real tissue. Although our proof-of-concept work relies on just six sections of healthy mouse kidney—hence on a single tissue type and species—it nonetheless shows that histological images can be synthesized directly from GEX data, opening new avenues for systematic exploration of gene–morphology links. As larger and more diverse ST datasets emerge, this approach could evolve into a powerful tool for pinpointing molecular drivers of early disease and for in silico simulations of the structural consequences of gene perturbations.

**Acknowledgments and Disclosure of Funding**

We thank the anonymous reviewers for their valuable comments. The work of F.L. was conducted as part of an internship at F. Hoffmann-La Roche Ltd. A.V. and J.V. are employees of F. Hoffmann-La Roche Ltd. This work was supported by the Roche Postdoctoral Fellowship to J.V.

## Appendix A. Training Details

### A.1 CLIP

The contrastive image-gene encoder is pretrained on a subset of 10000 image-gene spot pairs drawn uniformly at random from the VisiumHD training sample. We train for a total of 4200 iterations at a learning rate of $5e - 6$ on a NVIDIA A100-SXM4 80GB GPU. A batch size of $N = 128$ and a dropout rate of $p = 0.8$ is used. Validation of model convergence is performed on sample 1.

### A.2 StyleGAN-G

StyleGAN-G is trained on all available data pairs from the VisiumHD sample at a learning rate of $2e - 3$. Training is conducted in 2 progressive growing steps on 2 NVIDIA A100-SXM4 80GB GPUs. First, we train at $64 \times 64$ resolution for 16,000 iterations. For subsequent training at $128 \times 128$, we keep all layers up to $64 \times 64$ resolution fixed and train for 50,000 iterations. We use a total batch size of 128 and a per-GPU batch size of 8. The guidance strength was set to $\lambda = 0.2$.

### A.3 Stable Diffusion

The Stable Diffusion Pipeline was fine-tuned for 55 epochs on a NVIDIA A100-SXM4 80GB GPU, keeping all components except the U-Net frozen. We use the HuggingFace implementation of SD and fine-tune at a batch size of 32 and a learning rate of $1e - 5$. To enable classifier-free guidance, the GEX embedding prompt is set to the embedding of the vector of all zeroes with probability $p_{\text{uncond}} = 0.1$. We use the definition of classifier-free guidance reported in Saharia et al. (2022), where $w = 1$ corresponds to using no guidance.

## Appendix B. Data Details

The Visium HD processed training sample and the standard Visium processed sample 1 were sourced from 10XGenomics (2024, 2021) and represent the coronal section of a kidney. The remaining standard Visium processed samples 2-5 originate from a proprietary dataset and represent the transverse section of a kidney. Samples 1-5 yield 3124, 1490, 1670, 1421, and 1427 data pairs respectively. In Visium HD data, the location of the center $2\mu m$ bin is used as the spatial location of the pseudo-spot. The stain normalisation is applied to each WSI individually before extracting image patches.

## Appendix C. Real Image Data

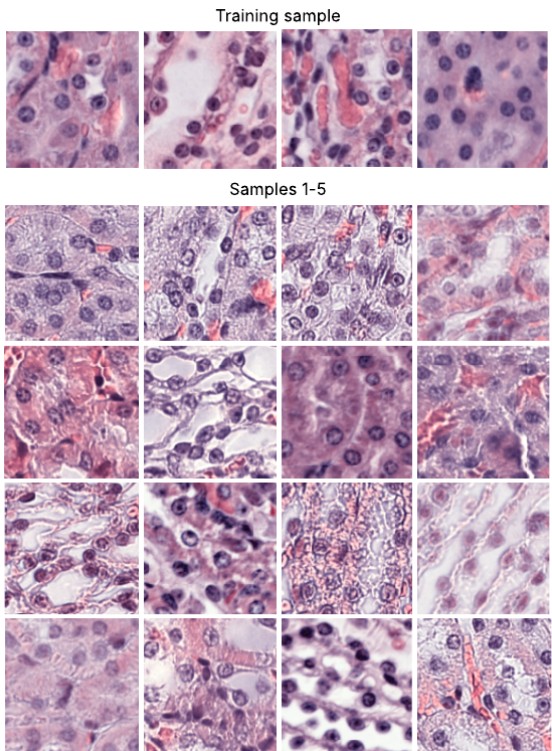

**Figure C.1.** Representative example patches from the training data (top row) and samples 1-5 (bottom)

## Appendix D. Image Embedding Space

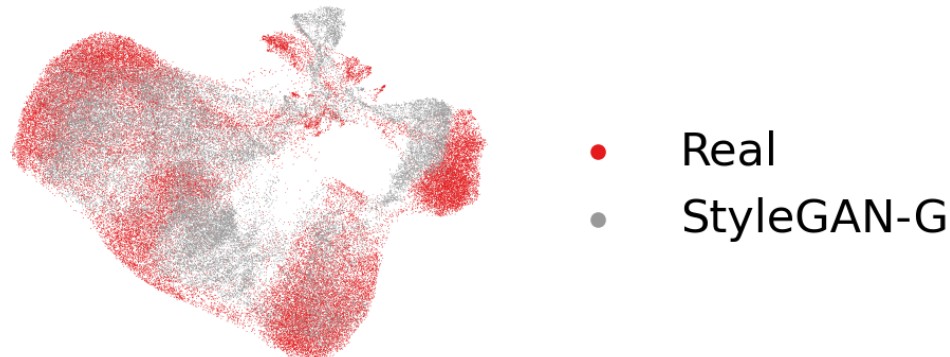

**Figure D.1.** UMAP of CLIP image embedding space

Figure D.1 displays a UMAP of the image embedding space generated using our CLIP model on both synthetic and real images (see also Tab. 1). It demonstrates that synthetic and real images are

well mixed and thus further strengthens the claim that the generated images accurately reflect the true data distribution.

## Appendix E. ARI Reference Clusters

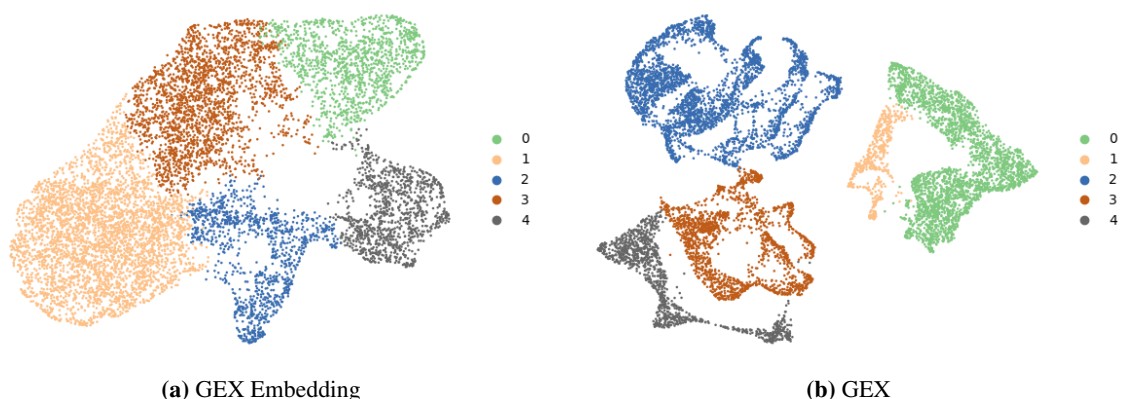

**(a)** GEX Embedding          **(b)** GEX

**Figure E.1.** Clusters found by clustering jointly across samples in (a) gene expression embedding space (b) gene expression space. Used as reference to compute Adjusted Rand Index.

## Appendix F. Significant GO:CC Terms

**Table F.1.** Significantly enriched GO:CC terms in sample 2 with more than 4 leading genes

| Term | NES ↑ | Query Cluster | Reference Cluster |
|---|---|---|---|
| Intracellular Membrane-Bounded Organelle | 1.852065 | Inner Medulla & Papilla | Cortex |
| Basolateral Plasma Membrane | 1.631743 | Inner Medulla & Papilla | Brown Adipose Tissue |
| Basolateral Plasma Membrane | 1.661647 | Cortex | OSOM |
| Endoplasmatic Reticulum Lumen | 1.951187 | Inner Medulla & Papilla | OSOM |
| Intracellular Organelle Lumen | 1.729963 | Inner Medulla & Papilla | OSOM |

**Table F.2.** Significantly enriched GO:CC terms in sample 3 with more than 4 leading genes

| Term | NES ↑ | Query Cluster | Reference Cluster |
|---|---|---|---|
| Vesicle | -1.853656 | Brown Adipose Tissue | OSOM |
| Intracellular Organelle Lumen | 1.78486 | Brown Adipose Tissue | OSOM |
| Endoplasmatic Reticulum Lumen | 1.703005 | Inner Medulla & Papilla | OSOM |
| Basolateral Plasma Membrane | 1.918574 | Inner Medulla & Papilla | Cortex |
| Nucleus | 1.852569 | Inner Medulla & Papilla | Cortex |
| Intracellular Membrane-Bounded Organelle | 1.700012 | Inner Medulla & Papilla | Cortex |
| Basolateral Plasma Membrane | 1.701396 | Inner Medulla & Papilla | Brown Adipose Tissue |
| Nucleus | -1.687745 | ISOM | Inner Medulla & Papilla |

**Table F.3.** Significantly enriched GO:CC terms in sample 5 with more than 4 leading genes

| Term | NES ↑ | Query Cluster | Reference Cluster |
|---|---|---|---|
| Endoplasmatic Reticulum Lumen | 1.937943 | Inner Medulla & Papilla | OSOM |
| Intracellular Organelle Lumen | 1.72107 | Inner Medulla & Papilla | OSOM |
| Basolateral Plasma Membrane | 1.938538 | Inner Medulla & Papilla | Cortex |
| Nucleus | -1.677389 | ISOM | Inner Medulla & Papilla |

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
