# OpenReview forum: "From Gene Expression to Tissue Morphology: Can Generative Models Uncover the Link?"
_MICCAI.org/2025/Workshop/COMPAYL — COMPAYL 2025_

### Official Review · Reviewer_sW3x · 2025-07-07
**From Gene Expression to Tissue Morphology: Can Generative Models Uncover the Link?**

**Rating:** 5
**Confidence:** 4

**Review:**

Summary
This paper presents a novel method for generating tissue morphology images directly from gene expression data.

Strengths
The core strength of this work lies in its two-stage strategic approach to synthesizing tissue morphology from gene expression profiles. The authors have designed a comprehensive set of experiments that effectively demonstrate both the quality of the generated images and their biological relevance. The results provide compelling evidence that gene expression data indeed contains information guiding tissue development and structure.

Weaknesses
One weakness is the limited comparison with other existing tissue generation models.
Also as all other papers in this area, the paper fails to address key questions central to this research area: Can the generated images accurately reflect or prognose specific diseases? And can this generative model explain the intricate link between gene expression and tissue morphology in a way that is readily understandable to humans? But as I metion, these are unsolved questions, the paper did a good try on the questions.

---

### Official Review · Reviewer_jUqK · 2025-07-16
**From Gene Expression to Tissue Morphology: Can Generative Models Uncover the Link?**

**Rating:** 4
**Confidence:** 5

**Review:**

This is an interesting paper evaluating how spatial transcriptomics can be used to inform on the morphology of tissues. To do so, Visium data are generated from tissues and synthetic images are produced and then compared to the ground truth / original H&E slides.
Perhaps the following could help to make the paper even stronger:
-	The images shown in Fig 2 are all tiles containing only 1 cell type. Very often it is the case with Visium that multiple cell types are picked up in one spot. It would be more convincing to see that the synthetic images can generate more than 1 cell type / tile
-	Of course the tiles are very small, but how do they look like when stitched spatially back together. It would be nice to see the larger WSI of the synthetic images side by side with the original.
-	Would it not be more informative to use single cell spatial? Or is the advantage here that the >18’000 gene transcripts are analysed?
-	Being able to identify subtle changes in morphology as a consequence of the GEX data  is an ambitious goal, and it would be more convincing to see even just one example of such a difference, especially since protein and post-translational changes are not analysed by Visium